# Tumor microbiome diffluences papillary thyroid cancer invasion

Lijuan Yuan [1,2], Ping Yang[1,2], Gang Wei[1], Xi'e Hu [1], Songhao Chen[1], Jianguo Lu[1], Lin Yang[1], Xianli He[1✉] & Guoqiang Bao [1✉]

Papillary thyroid carcinoma (PTC) has a high incidence, and its proper treatment remains challenging. Therefore, identifying PTC progression markers is essential. Here, using 16S RNA sequences, we analyzed the PTC intratumor microbiome and its role in tumor progression. Substantial microbial abundance was detected in PTC from all patients. The tumor bacterial diversity in patients with advanced lesions (T3/T4) was significantly higher than that in patients with relatively mild lesions (T1/T2). Importantly, we identified signatures of eight tumor bacterial taxa highly predictive of PTC invasion status. Hence, microbial host factors—independent of the genomic composition of the tumor—may determine tumor behaviors and patient outcomes. Furthermore, the correlation between specific bacterial genus and thyroid hormones or autoimmune thyroid disease-related antibodies may indicate the potential contribution of the microbiome in the relationship between autoimmune thyroid disease or irregular thyroid function and PTC progression, intervention of which might therefore be worth exploring for advancing oncology care.

[1] Department of General Surgery, Tangdu Hospital, The Air Force Military Medical University, Xi'an, China. [2]These authors contributed equally: Lijuan Yuan, Ping Yang. ✉email: wanghe@fmmu.edu.cn; guoqiang@fmmu.edu.cn

Thyroid cancer (TC) is the most common malignancy of endocrine organs[1]; recently, its incidence has rapidly increased compared with other solid tumors. Papillary thyroid carcinoma (PTC) is the most common type of TC. Most patients with PTC have a 5-year survival rate of over 95%; however, more than 3–5% of patients still experience recurrent metastatic disease[2], indicating a poor prognosis. The main challenge in treating PTC is the properties that can destroy neoplastic foci and inhibit the natural course of the disease[3]. Evidence shows that activating somatic alterations in genes may contribute to thyroid tumor development and progression[4]. Recently, altered compositions of intratumor microbiomes have been observed in different cancers[5], which has been indicated to influence tumor behaviors and patient outcomes independent of the tumor genomic composition[6].

Recent studies have reported that alterations in commensal microorganisms, comprising bacteria, fungi, protozoa, and viruses, are strongly associated with oncogenesis and tumor progression[7]. The bacterial microbiome has various metabolic functions, beyond those observed in the host genome[8]. Substantial evidence suggests that the gut microbiome may confer susceptibility to certain cancers, alter interactions with the human immune system[9], and control cancer response to anticancer therapy by modulating the tumor immune microenvironment[10–12]. Furthermore, appropriate microbiome modulation may ameliorate the antitumor effects of PD-1 blockade[13], indicating the potential role of the microbiome in aiding therapeutic agents.

The key role of the gut microbiome as a host factor mediating thyroid function has also been highlighted. Germ-free rats, raised in sterile conditions and lacking gut bacteria completely, have smaller thyroid glands than conventionally raised rats, suggesting a crucial role of these microbes in thyroid health[14]. Intestinal imbalances have been attributed to low thyroid function and thyroid autoimmunity[15]. Both TC and thyroid nodules are associated with the composition of the gut microbiome[16]. The gut microbiome of participants with high-grade thyroid nodules is characterized by greater amino acid degradation and lower butyrate production[17], suggesting that the link between gut bacteria and thyroid disorders may depend on metabolite-mediated interactions in both host and microbes. In addition, gut bacteria support the conversion of the thyroid hormone thyroxine (T4) to triiodothyronine (T3) in the intestine and modulation of both T helper 1 (Th1) and Th2 immune responses[18]. Microbes might influence thyroid hormone levels by regulating iodine cycling, and they also influence interactions between the host and microbiome[15].

However, the role of a specific intratumor microbiome is unclear because of its low biomass. Intratumor microbiomes have been revealed in cancers, such as head and neck squamous cell carcinoma[19], cholangiocarcinoma[20], as well as colorectal[21], pancreatic[6], lung[22], urothelial[23], cervical[24], and breast cancer[25]. Moreover, each tumor type has a distinct microbiome composition[5]. Furthermore, intratumor microbiome composition is reportedly associated with the gut microbiome in pancreatic adenocarcinoma (PDAC) and influences the host immune response and natural history of the disease[6]. It has further been proven that tumor-resident intracellular bacteria promote metastasis in breast cancer[26].

The link between TC and intratumor microorganisms has not been well delineated. In addition, the composition of the human intra-PTC microbiome that may contribute to the natural history of PTC warrants further research. To gain a more comprehensive understanding of host-microbiome interactions, bacteria in TC tissues were analyzed using 16S RNA sequencing in the present study[5,6].

## Results

**Tumor microbiome communities are significantly associated with tumor invasion in patients with resected PTC**. The demographic and clinical characteristics of the patients with PTC are shown in Supplementary Tables S1 and S2. The tumor microbial diversity was compared among surgically resected patients in different clinical stages. The tumor microbial diversity was measured using different methodologies (Sobs, Shannon, and Simpson indices). The alpha-diversity ($\alpha$-diversity) of the tumor microbiome was significantly lower in patients with T1/T2 PTC than in those with T3/T4 PTC, as shown by the Shannon and Simpson indices ($p = 0.0309$ and $p = 0.0088$; Wilcoxon rank-sum test was also performed Fig. 1a). This indicated lower microbiome diversity (Shannon and Simpson indices) in patients with T1/T2 PTC. Microbiome richness was measured by the number of observed OTUs (Sobs index), and no significant differences were found among different clinical stages (Fig. 1b). To gain a better understanding of the role of microbiome diversity, beta-diversity ($\beta$-diversity) was used to carry out a principal coordinate analysis (PCoA) using Bray–Curtis metric distances (Fig. 1c; $p = 0.001$ [T1/T2/T3/T4]; $p = 0.001$ [T1_2/T3_4]). Significant differences in $\beta$-diversity were observed, further suggesting that the tumor microbial communities varied during tumor progression.

The general landscape of the tumor microbiome composition was assessed by considering the relationship between PTC intratumoral bacterial diversity and clinical stages (Fig. 2). Enrichment of specific bacterial communities was evaluated at different taxonomic levels (Supplementary Fig. S1). At the genus level, *Pseudomonas* was the dominant bacterium, followed by *Rhodococcus*, *Ralstonia*, *Acinetobacter*, and *Sphingomonas*. Next, we sought to determine whether tumor microbiome composition differed among patients with PTC at stages T1, T2, T3, and T4. We found similar communities among patients with PTC at various stages (Fig. 2b, c, Supplementary Data 1–3, 8). However, considerable differences in the percentage of community abundance were also observed via Circos analysis (Fig. 2d, e), showing the percentage of the dominant genera in distinct groups. For example, the percentage of *Pseudomonas* was higher in T1 or T2 than in T3 or T4.

To further investigate the specific changes in the microbiome in the tumors of patients with PTC at different stages, the relative abundance of taxa was assessed. At the genus level, three genera, namely *Pseudomonas* ($p = 0.0017$), *Rhodococcus* ($p = 0.02644$), and *Sphingomonas* ($p = 0.0073$), displayed a difference in abundance among various stages (Fig. 2f and Supplementary Fig. S2a). *Pseudomonas* spp., the most abundant genus in all groups, presented a higher abundance in tumors of patients with T1 and T2 PTC than in those with T3 or T4 PTC ($p = 0.0049$, $p = 0.0138$, $p = 0.010$, and $p = 0.0028$, respectively; Fig. 2f). *Rhodococcus* abundance was also significantly higher in patients with T1 PTC than in those with T3 PTC ($p = 0.0032$; Fig. 2f), and *Sphingomonas* was more abundant in T1 and T2 than in T3 ($p = 0.0001$ and $p = 0.0005$, respectively; Fig. 2f, Supplementary Data 4). To further investigate these findings, we conducted high-dimensional comparisons using linear discriminant analysis of effect size (LEfSe). We detected considerable differences in the predominance of bacterial communities between T1_2 and T3_4 (Fig. 2g, h Supplementary Data 5). The T1_2 tumors exhibited a predominance of *Pseudomonas* (Pseudomonadales at the order level), *Rhodococcus* (Corynebacteriales at the order level), and *Sphingomonas* (Sphingomonadales at the order level) at the genus level. In contrast, T3_4 tumors were dominated by Streptococcus, Granulicatella, Haemophilus g_unclassified_o_Rhizobiales, and g_norank_f_norank_o_-Coriobacteriales at the genus level (Fig. 2g, h). We then used the eight genera for the area under the curve (AUC)-receiver operating characteristic (ROC) analysis. The combination of the eight taxa (*Pseudomonas*, *Rhodococcus*, and *Sphingomonas Streptococcus*,

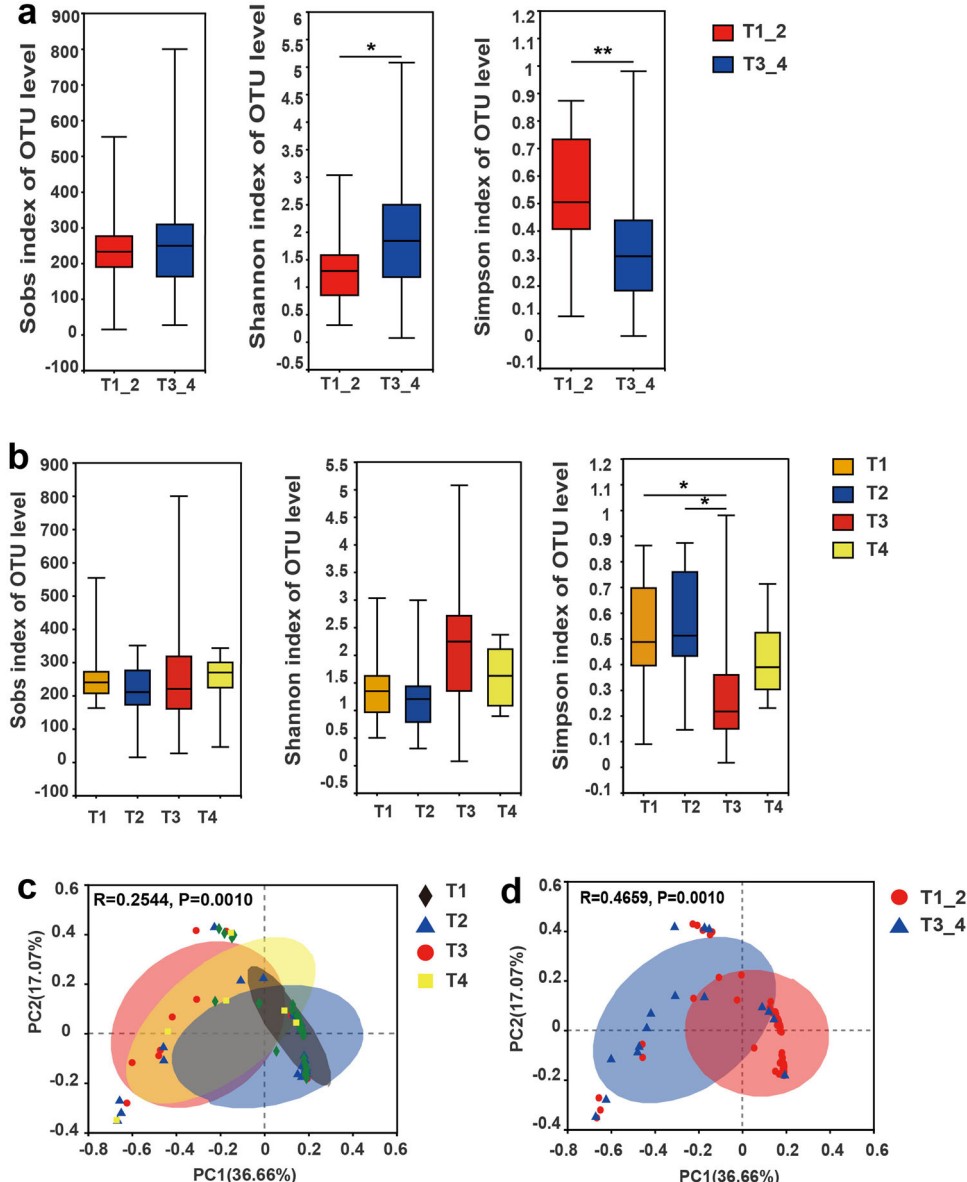

**Fig. 1 Changes in the intratumor microbiomes of patients with papillary thyroid carcinoma (PTC) at different clinical stages. a** Alpha-diversity (α-diversity) at the operational taxonomic unit (OTU) level (estimated using Sobs, Shannon, and Simpson indices) in patients with PTC in groups T1_2 and T3_4. **b** α-Diversity at the OTU level (estimated using Sobs, Shannon, and Simpson indices) in patients with PTC of clinical stages T1, T2, T3, and T4 (*$p < 0.05$ and **$p < 0.01$). Note: The Sobs values indicate the number of observed OTUs, Shannon diversity index, and Simpson diversity index for an OTU definition, respectively. The lower the value of the Simpson index, the higher the diversity (opposite of the Shannon index). Error bars indicate SD. **c** Principal coordinate analysis (PCoA) score plots based on the Bray–Curtis distance at the OTU level in patients with PTC in groups T1_2 and T3_4. **d** PCoA score plots based on the Bray–Curtis distance at the OTU level in patients with PTC at clinical stages T1, T2, T3, and T4. Wilcoxon rank-sum test and analysis of similarities (ANOSIM) were used for the analysis of intragroup differences in α-diversity and PCoA analysis, respectively.

*Granulicatella, Haemophilus* g_unclassified_o_Rhizobiales, and g_norank_f_norank_o_Coriobacteriales) resulted in an AUC of 0.83 in the T1_2 and T3_4 groups (Fig. 2i), and 0.91 in the T1 and T4 groups (Supplementary Fig. S2b), which showed higher discriminative capacity than in the group of bacteria filtered by random forest analysis (Supplementary Fig. S3). In addition, we conducted an AUC-ROC analysis with a separate set of samples using the combination of the eight taxa, which resulted in an AUC of 0.79 for TT1_2 and T3_4 groups (Supplementary Fig. S5b), although the composition of bacterial communities was evaluated at different taxonomic levels (Supplementary Fig. S5a). This confirmed the potential of the eight-genera microbiome signature to discriminate and influence PTC invasion status.

**Intratumor microbial dysbiosis is related to thyroid function.** Identified functional predictions differentially presented meta-Cyc pathways between patients with T1_2 and T3_4 PTC. The top pathways that differed between these two groups were listed according to the effect size (difference between proportions >0.2 and the ratio of proportions >10; $p < 0.05$; Supplementary Fig. S2c). The T1_2 group had higher proportions of most of the pathways, including fatty acid salvage, sulfate reduction (assimilatory), fatty acid and beta-oxidation I, octane oxidation, and ʟ-tyrosine degradation. While the T1_2 group had higher proportions of super pathways of purine nucleotide de novo biosynthesis and palmitate biosynthesis II (bacteria and plants).

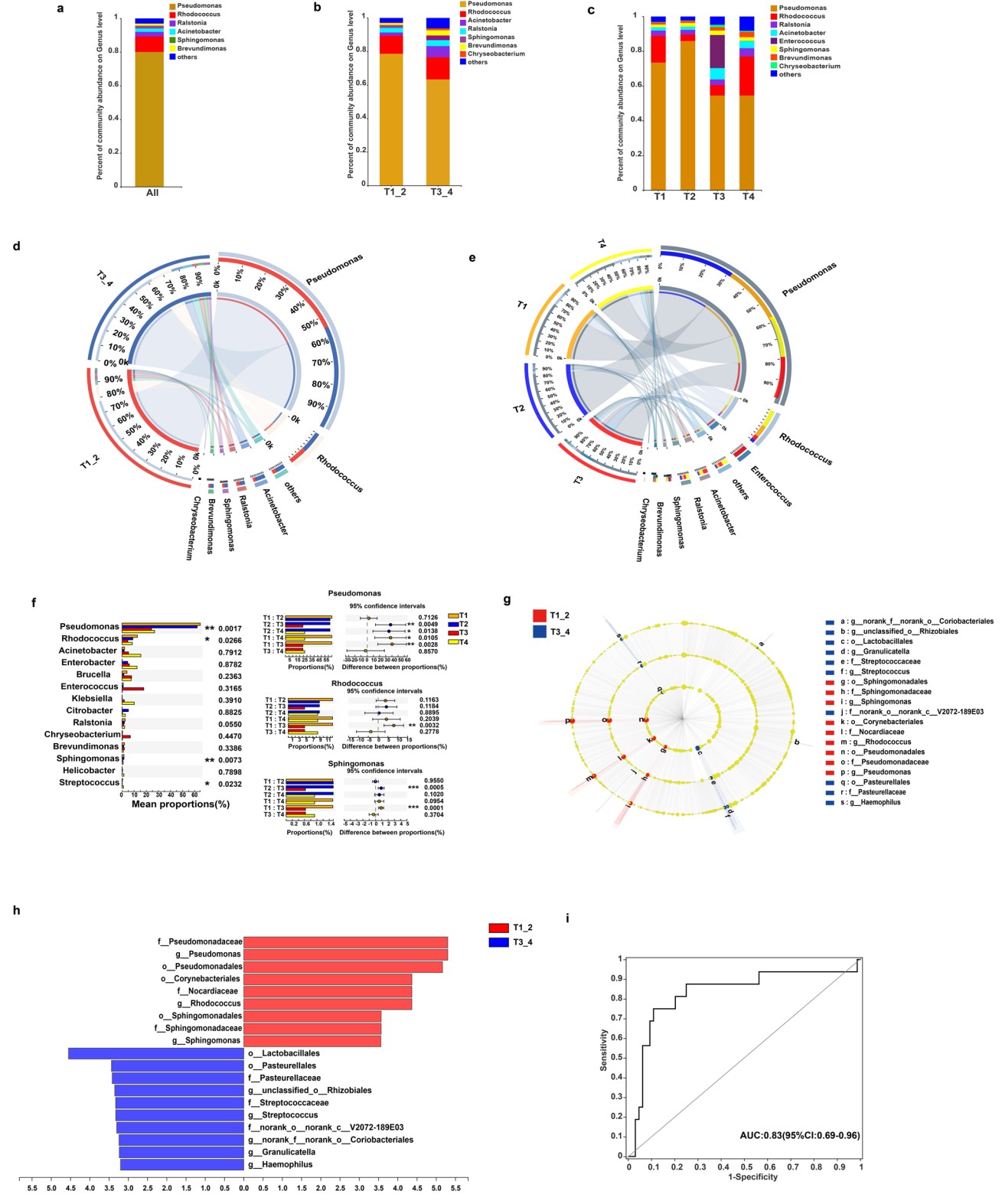

Furthermore, the association among the thyroid-related hormones free T4 (FT4), T4, FT3, T3, and thyroid-stimulating hormone (TSH) and microbial abundance were assessed. The levels of different thyroid hormones were related to different microbial genera (Fig. 3a and Supplementary Fig. S4a, Supplementary Data 6). A positive relationship was observed between FT4 and *Neisseria* and norank_f__norank_o__Chloroplast, FT3,

and *Treponema*, whereas a negative relationship was observed between FT4 and *Klebsiella*, T4 and *Klebsiella* and *Escherichia-Shigella*, T3 and *Granulicatella*, and TSH and norank_f__ norank_o__Clostridia_UCG-014 and *Prevotella*. In addition, positive correlations between FT4 and norank_f__norank_ o__Clostridia_UCG-014 and negative correlations between hormones T3 and FT3 and *Granulicatella* were found in the

**Fig. 2 Tumor microbiome communities significantly differ among clinical stages. a–c** Bar plots of the genus level in patients with PTC. **a** Relative abundance for all patients with PTC. The genus level shows a more refined taxonomic level than the phylum level in bacteria and can help find particular species. **b** Relative abundance for patients with PTC in groups T1_2 and T3_4. **c** Relative abundance for patients with PTC in clinical stages T1–T4. **d** and **e** Distribution of microbial communities in each group at the genus level. **d** Distribution of microbial communities for patients with PTC in groups T1_2 and T3_4. **e** Distribution of microbial communities for patients with PTC at clinical stages T1–T4. The data are visualized using Circos. The width of the bars for each genus indicates the relative abundance of that genus in the group. **f** Microbiome alterations at the genus level in patients with PTC of clinical stages T1–T4. Intragroup differences were analyzed using the Kruskal–Wallis test, and differences between groups were analyzed based on the post-hoc test using Welch's uncorrected test, adjusted by false discovery rate. The top three differential bacteria (genus) identified were tested individually. **g** Taxonomic cladogram from LEfSe, depicting taxonomic associations between microbiome communities from patients with T1_2 and T3_4 PTC. Each node represents a specific taxonomic type. Yellow nodes denote taxonomic features that are not significantly differentiated between T1_2 and T3_4. Red nodes denote taxonomic types more abundant in T1_2 than in T3_4, whereas blue nodes represent taxonomic types more abundant in T3_4. **h** LDA score computed from differentially abundant features between T1_2 and T3_4. The criterion for feature selection was log(LDA score) >3.2. **i** Receiver operating characteristic (ROC) curve of the relative abundance of taxa as a predictor of clinical T stage status. The eight differential bacteria (genus) were tested in aggregate as average. The confidence interval threshold was set as 0.95 ($p = 0.00003$).

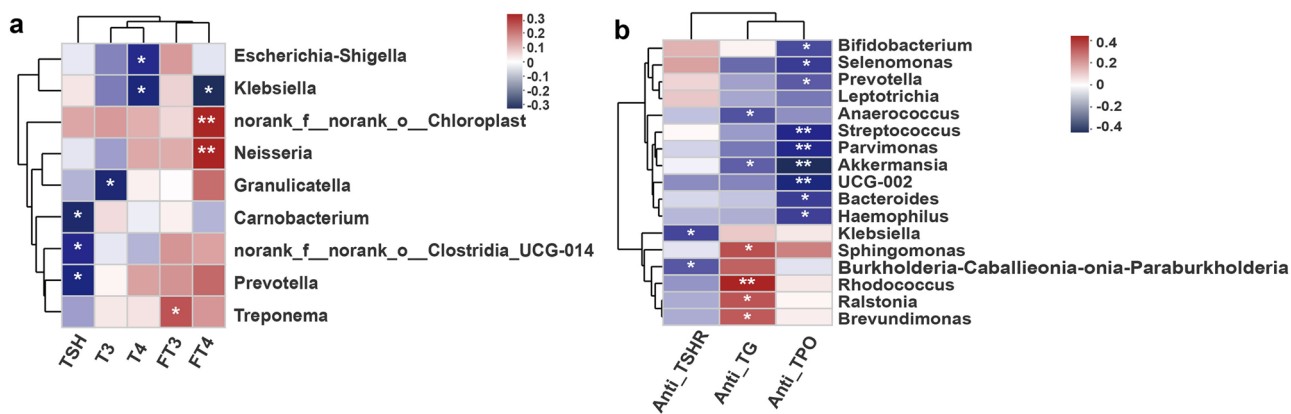

**Fig. 3 Heatmap of Spearman's correlation analysis between the specific PTC intratumor microbiome and the clinical factors. a** Heatmap of Spearman's correlation analysis between the specific bacterial and thyroid-related hormones. *$p < 0.05$ and **$p < 0.01$. Heatmap color keys indicate Spearman correlation coefficient. **b** Heatmap of Spearman's correlation analysis between the specific bacterial and thyroid diseases (AITD)-related antibodies. *$p < 0.05$ and **$p < 0.01$. Heatmap color keys indicate Spearman correlation coefficient.

other set of samples (Supplementary Fig. S5c). However, all these microbes showed a relatively lower intratumor richness in PTC.

The association between the intratumor microbiome and autoimmune thyroid disease (AITD)-related antibodies was also investigated, as AITD affects the entire metabolism in the human body, and the immune status may also contribute to PTC progression. Autoantibodies against thyroid peroxidase (TPO) and thyroglobulin (TG), which characterize Hashimoto's thyroiditis (HT), and thyroid-stimulating receptors (TSHR), a marker for Graves' disease (GD), were used in this study. The relationship between microbial abundance and the levels of the three autoantibodies was assessed (Fig. 3b, Supplementary Data 7 and Supplementary Fig. S4b). The levels of anti-TSHR, anti-TG, and anti-TPO antibodies were adjusted; values <0.3 were set as 0.2, values <10 were set as 9, and values <5 were set as 4. A negative correlation was found between anti-TSHR levels and *Klebsiella* and *Burkholderia-Caballeronia-Paraburkholderia*. Meanwhile, the anti-TG levels positively correlated with *Sphingomonas*, *Rhodococcus*, *Ralstonia*, and *Brevundimonas* but negatively correlated with *Anaerococcus* and *Akkermansia*. Nine genera, namely UCG-002, *Streptococcus*, *Parvimonas*, *Akkermansia*, *Bacteroides*, *Haemophilus*, *Selenomonas*, *Prevotella*, and *Bifidobacterium*, exhibited a negative relationship with the anti-TPO levels.

Correlation analysis between the abundance of these microbes and AITD-related antibodies was also conducted in the other set of samples (Supplementary Fig. S5c). *Bifidobacterium* and

*Haemophilus* abundance showed a positive relationship with the anti-TPO levels, while *Leptotrichia* abundance showed a positive relationship with the anti-TSHR levels. This further proved the correlation between the abundance of microbes and thyroid-related hormones and AITD-related antibodies, while the interaction may differ according to individual conditions. Functional predictions identified different pathways for these specific microbes between patients with T1_2 and T3_4 PTC (Supplementary Fig. S4c). These metabolite differences may disrupt thyroid hormones and AITD-related antibody levels related to thyroid function.

**Associations between clinical variables and the intratumor microbiome.** An association between sex and intratumor microbiome diversity was found in patients with PTC (Fig. 4). A higher microbiome diversity (alpha-diversity (α-diversity)) was observed in females (Fig. 4a), who are expected to have a higher PTC incidence. However, no obvious difference in beta diversity was found ($p = 0.3000$). Of note, *Rhodococcus*, *Ralstonia*, *Chryseobacterium*, and *Burkholderia-Caballeronia-Paraburkholderia* were more abundant in female patients with PTC than in male patients (uncorrected $p = 0.0413$, $p = 0.0092$, $p = 0.0275$, and $p = 0.0008$, respectively, Wilcoxon rank-sum test; Fig. 4c). Remarkably, the sex-associated genus was the same T1/T2 PTC-enriched genus *Rhodococcus*. Moreover, a higher PTC microbiome diversity might have a worse association with PTC. We also noted correlations between intratumor bacteria or their

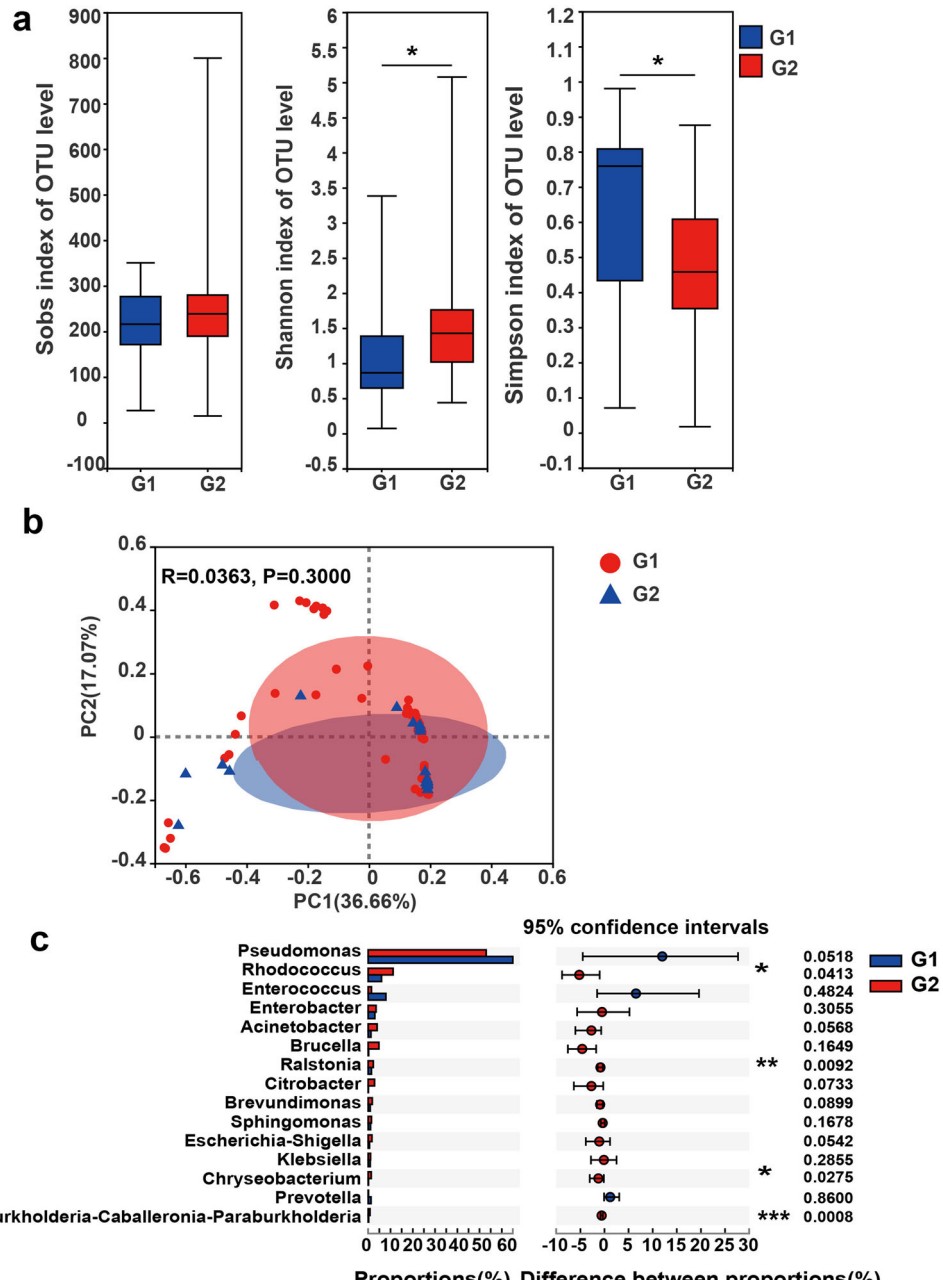

**Fig. 4 Changes in the intratumor microbiomes of patients with PTC in different sexes. a** α-Diversity at the OTU level (estimated using Sobs, Shannon, and Simpson indices) in patients of different sexes with PTC (*$p < 0.05$ and **$p < 0.01$). Error bars indicate SD. **b** PCoA score plots based on the Bray–Curtis distance at the OTU level in patients of different sexes with PTC. Wilcoxon rank-sum test and ANOSIM were used for the analysis of intragroup differences in α-diversity, Simpson index, and PCoA analysis, respectively. **c** Microbiome alterations at the genus level in patients of different sexes with PTC. Differences between groups were analyzed using the Wilcoxon rank-sum test, and the 95% confidence interval (CI) was calculated using the bootstrap method. G1: male; G2: female.

predicted functions in patients with different tumor subtypes, ages, or lymphatic metastasis status. However, there was no significant difference with respect to the α- and β-diversities (Supplementary Fig. S6).

## Discussion
In this study, we explored the role of intratumor microbiome composition in PTC. Overall, we observed a substantial microbiome abundance in the PTCs of all patients. We found that patients with advanced lesions (T3 or T4) had a significantly higher tumor bacterial diversity (α-diversity) than those with relatively mild lesions (T1 or T2). Furthermore, the T1/T2 and T3/T4 groups had distinctive tumor microbiome signatures with specific bacterial genera that may indicate PTC invasion status.

Recent studies have shown that the microbiome composition is distinct in each tumor type[5] and that the intratumor microbiome composition affects patient outcomes[6], responses to cancer therapies[27], and play an important role in promoting cancer metastasis[28]. Our findings suggest an association between intratumor microbiome diversity, composition, and tumor invasion. We found a higher α-diversity in the tumor microbiome of patients with PTC in stages T3 and T4 than in T1 and T2. We identified a signature consisting of eight tumor bacterial taxa that

were highly predictive of PTC invasion status. A recent study demonstrated that the re-examination of whole-transcriptome RNA-sequencing (level 3 normalized mRNA expression read counts) and 450k DNA methylation sequencing data from the Genomic Data Commons legacy archive for microbial sequences enabled the identification of the association between different PTC subtypes and microbial abundance[29]. Extending our analysis to the functional level demonstrated the correlation of certain immune or genetic traits and bacteria, despite a very large variation at the taxonomic level. In this study, using 16S sequencing methods, we validated the presence of bacteria in PTC and their function in tumor progression. These data suggest that PTC microbial host factors, independent of the genomic composition of the tumor, may determine tumor behavior and patient outcomes.

Most bacterial communities found in the tumoral milieu are commonly present in the gut microbiome[30], suggesting that bacterial translocation from the gut to the tumor in other sites might occur. The crosstalk between the PDAC microbiome composition and gut microbiome has been demonstrated by Riquelme et al.[6]. Four bacterial phyla are predominant in the PTC intratumor, namely, *Proteobacteria*, *Actinomycetes*, *Firmicutes*, and *Bacteroides* (Supplementary Fig. S1), which have also been reported as the main strains in the gut[31]. However, when the abundance was set to more than 0.01, different dominant strains were found in the PTC intratumor microbiome at the genus level compared with the gut microbiome. For instance, *Pseudomonas*, *Rhodococcus*, *Ralstonia*, *Acinetobacter*, *Sphingomonas*, and *Brevundimonas* were only found in PTC but not in the gut (even in the gut of patients with TC) as previously reported[8,16], which may account for the change in the environment of the microbiome. Nevertheless, the translocation of bacteria from the gut to PTC still needs further exploration. Bacterial transplantation in an animal model or function research in vitro may represent a suitable approach.

It is known that thyroid hormones could indicate thyroid function, and TSH and FT3 levels are higher in TC cases than in normal ones. The gut microbes influence thyroid hormone levels by regulating iodine uptake and degradation, and enterohepatic cycling[32]. Intratumor microbes have also been shown to be related to thyroid hormones. It has been reported that intratumoral bacteria are metabolically active; bacteria often found within tumors can alter the chemical structure of common chemotherapeutic agents, changing their activity and consequently their effective local concentration[33]. The environment formed by thyroid hormones also affects microbial abundance and activity. Therefore, the interaction between intratumoral bacteria and thyroid hormone levels may contribute to tumor progression and invasion.

The effect of the tumor microbiome on AITD-related antibodies was also observed in this study. The gut microbiome is reportedly involved in HT and GD[15], which are both AITDs and major causes of hypothyroidism and hyperthyroidism, respectively. However, its role in the AITD response is not entirely clear. Studies in humans have reported a higher abundance of *Prevotellaceae* and *Pasteurellaceae* in patients with Graves' disease[34]. A decrease in the levels of *Bifidobacteria* and *Lactobacillaceae* has also been reported in patients with hyperthyroidism[35]. In patients with HT, a significant increase in the abundance of *Bacteroides* species and a decrease in that of *Bifidobacterium* in stool samples have been observed[36]; however, a decreased abundance of *Bacteroides* members has also been observed[37]. *Prevotellaceae*, *Bacteroides*, and *Bifidobacteria* also showed a negative relationship with anti-TPO levels in this study, further indicating their immunoregulatory effect. Previous studies have shown that *Lactobacillus* spp. and *Bifidobacterium* spp. may induce antibodies to

cross-react with thyroperoxidase and thyroglobulin, owing to their molecular mimicry[38]. However, their roles in HT and GD remain to be explored. Furthermore, the mechanism by which the tumor microbiome modulates or resets the immune system may improve or impair the immune response against the tumor and affect PTC behavior and outcome.

Our study had several limitations. First, the exact mechanism of the PTC microbiome needs to be clarified further, even though a correlation between the intratumor microbiome and tumor invasion was observed. Second, the association between the microbiome and environmental factors, such as drinking, smoking, and dietary habits, was not accounted for. Third, easier-to-obtain oral microbiota may exhibit an association with cancer. Future studies including a comparison of the intratumoral and oral microbiome may facilitate the characterization of the relationship between the tumor microbiome and tumor status.

In conclusion, the composition of the PTC microbiome and its association with tumor invasion were determined in this study. The tumor microbiome may interact with the thyroid hormone and autoimmune antibodies to regulate the tumor microenvironment, which may contribute to tumor invasion. Significant differences in microbiome diversity were observed between female patients with PTC and male patients with PTC, and between female patients with T3/T4 PTC and male patients with T1/T2 PTC, suggesting microbiome characteristics in patients with traits such as high PTC incidence (female patients) and advanced PTC (T3/T4 PTC). Our findings indicate that tumor-resident microbiota, albeit their low biomass, play a key role in promoting PTC progression, which might be worth exploring to advance oncology care.

## Methods

**Study cohort**. This study was approved by the Institutional Review Board and Human Ethics Committee of the Air Force Military Medical University (Xi'an, China). Tumor samples from 80 patients with PTC were obtained in Tangdu Hospital from 2013 to 2018. All patients with PTC were newly diagnosed and histologically confirmed to have PTC, and other types of TC were excluded. They were included in the study, and all participants provided written informed consent. Demographic and clinical variables were collected during clinic visits. T stage and lymphatic metastasis status (N) of the patients were considered, which are represented in the TNM stages: T indicates the size of the main (primary) tumor and whether it has metastasized into nearby areas; N describes the extent of spread to nearby (regional) lymph nodes; M indicates whether cancer has spread (metastasized) to other organs of the body.

**Validation set of the study cohort**. For further validation of the eight-genera microbiome signature that could reliably distinguish between different T stages, we recruited another set of patients from Tangdu Hospital in November 2021. All patients were diagnosed with T1 or T2 PTC (Supplementary Tables S3 and S4). Therefore, a ROC analysis was conducted using this set of patients and the patients with T3 or T4 PTC in the old set of samples. In addition, the differential abundance of microbes and their correlations to thyroid hormones and autoimmune thyroid disease-related antibodies were also investigated in this set of samples.

**Sample collection and DNA extraction**. The tissue samples were cut into 0.3-cm sections, quick-frozen in liquid nitrogen, and stored at −80 °C. DNA was extracted using the E.Z.N.A.® soil DNA Kit (Omega Bio-tek, Norcross, GA, USA) according to the manufacturer's instructions. The concentration of bacterial DNA was measured using a Nanodrop 2000 spectrophotometer (Thermo Fisher Scientific, Waltham, MA, USA).

**16S rRNA gene sequencing**. The hypervariable regions V3-V4 of the bacterial 16S rRNA gene[5,6] were amplified with primer pairs *338F* (5′-ACTCCTACGGGAGG-CAGCAG-3′) and *806R* (5′-GGACTACHVGGGTWTCTAAT-3′) using an ABI GeneAmp® 9700 polymerase chain reaction (PCR) thermocycler (Applied Biosystems, CA, USA). The amplicons were purified using the AxyPrep DNA Gel Extraction Kit (Axygen Biosciences, Union City, CA, USA) according to the manufacturer's instructions and quantified using a Quantus™ Fluorometer (Promega, Madison, WI, USA). The purified amplicons were pooled in equimolar and paired-end sequenced (2 × 300) on an Illumina MiSeq platform (Illumina, San Diego, CA, USA) according to the standard protocols of Majorbio Bio-Pharm Technology Co., Ltd. (Shanghai, China). A negative control (including control

samples during sample collection and the following experiment) was subjected to DNA extraction and PCR to ensure that there was no contamination[5].

**Microbial analysis**. The data were analyzed using the free online Majorbio Cloud Platform (www.majorbio.com). The raw 16S rRNA gene sequencing reads were demultiplexed, quality-filtered using Trimmomatic, and merged using FLASH. Operational taxonomic units (OTUs) with a 97% similarity cutoff were clustered using UPARSE (version 7.1, http://drive5.com/uparse/), and chimeric sequences were identified and removed. The taxonomy of each OTU representative sequence was analyzed using RDP Classifier (http://rdp.cme.msu.edu/) against the 16S rRNA database (Silva SSU128) with a confidence threshold of 0.7. OTUs with several sequences < 0.005% of the total number of sequences were removed from the OTU table. After filtering, an average of 45,390 reads per sample was obtained (min: 29,987 and max: 81,902). In addition, rarefaction was performed on the OTU table to prevent methodological artifacts due to varying sequencing depths. Alpha-diversity (α-diversity) was measured based on species richness and diversity from the rarefied OTU table. Beta-diversity (β-diversity) was estimated by computing the Bray–Curtis distance and visualized with PCoA. To dissect possible species for OTUs, we performed a MegaBLAST search to align the reads of OTUs against reference sequences in the National Center for Biotechnology Information 16S rRNA database.

**Statistics and reproducibility**. Statistical analyses were performed using the free online Majorbio Cloud Platform. The Wilcoxon rank-sum test and analysis of similarities were used to analyze intragroup differences in the α-diversity Simpson index and PCoA analysis, respectively. For the comparison of continuous variables, the Mann–Whitney $U$ test (Kruskal–Wallis test for more than two groups) was used. The Spearman correlation test was used for correlation analysis. The relative abundance was arcsine square-root transformed. Intragroup differences were analyzed using the Kruskal–Wallis test. The differences between groups were analyzed by post-hoc tests using Welch's uncorrected test adjusted by false discovery rate. The differences between groups were analyzed by two-tailed Wilcoxon rank-sum tests. The 95% confidence interval was calculated using the bootstrap method (0.95). Circos analysis was used to describe the association between samples and species; LEfSe was used for linear discriminant analysis. To evaluate the discriminatory ability of the significant genus, a ROC curve was constructed, from which the AUC was calculated. The random forest model using cross-validation was also used to filtrate the significate genus, but a lower AUC of 0.72 was obtained (Supplementary Fig. S3). PICRUSt2 was used to predict the function. The differences between groups were analyzed using the two-sided Welch's t-test using the STAMP software. The clinical variables were analyzed by chi-square tests using SPSS 24.0 (IBM Corp., Armonk, NY, USA).

**Reporting summary**. Further information on research design is available in the Nature Research Reporting Summary linked to this article.

## Data availability

All data generated or analyzed during this study are included in this published article and its Supplementary information files. Sequencing data have been submitted to the NCBI Short Read Archive (SRA) database (https://www.ncbi.nlm.nih.gov/sra) under bioproject number Accession: PRJNA854334, Submission ID: SUB11718364. The processed data are available in the Supplementary Data.

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

## Acknowledgements

The authors thank Air Force Military Medical University for funding 2020ZTB11 and Tangdu Hospital for funding 2019JSYJ014. We acknowledge all staff from the department of general surgery in Tangdu Hospital. We thank Majorbio company for the 16S rRNA sequencing and analysis platform.

## Author contributions

G.B. and X.H. contributed to the conception of the study; L.Y. and P.Y. performed the experiment and made significant contributions to analysis and manuscript preparation; G.W., X.H., and S.C. performed the data analyses; J.L. and L.Y. helped perform the analysis with constructive discussions.

## Competing interests

The authors declare no competing interests.
