## [Peer Review File · Communications Biology]

Reviewers' comments:

Reviewer #1 (Remarks to the Author):

In the current manuscript, the authors present an analysis of the thyroid tumor microbiome and how it may affect carcinogenesis. The authors effectively present an interesting and concise conclusion of the relationship between the microbiome, TH, and antibodies and how they affect tumor invasion. However, there are some concerns with the justifications of certain methods and elaboration on their conclusions that must be addressed:

- The manuscript has formatting and grammatical errors. More thorough proofreading is necessary.
- It would be helpful to discuss examples of how microbes have shown to alter cancer progression or influence response to therapeutics in the Introduction. While the authors make it clear that the gut and intra-tumor microbiome are distinct in cancer samples, they should devote more time to explaining why and how microbes can be used as therapeutic agents.
- There are different subtypes of PTC: classical, follicular variant, and tall cell. Each of these subtypes have shown to have different prognoses and response to treatment. Did the authors separate the tissue samples based on subtype? If not, why not?
- Did the authors perform any procedures to correct for contamination? If not, the authors should discuss why not and how plausible/credible the results are without contamination correction.
- Since the study primarily presents correlational data between microbe abundance and tumor invasion, thyroid hormones, and autoimmune antibodies, the authors should address how future research can confirm causality or the mechanism by which the microbes act (if such experiments are outside the scope of this study).
- The author should not end the conclusion with an "In addition..." statement. The conclusion should do a better job in summarizing the study, synthesizing the results, and proposing next steps.

Reviewer #2 (Remarks to the Author):

In this study, the authors investigate how the tumor microbiome may determine tumor behaviour, affect patient outcome, and influence thyroid cancer. Interestingly, the intra-tumor dysbiosis, as well as its relationship with thyroid function, with regards to hormones and the immune system is also investigated. The topic is of interest to the readers of Communications Biology. Below are the reviewers' comments that authors should address to improve the quality of the publication.

This is absolutely an important point worth studying, nonetheless, the following paper has been already published regarding intra-tumor microbiota and thyroid cancer. I think the authors should further stress the novelty of their approach against this paper:

- Gnanasekar, A. et al. (2021). The intratumor microbiome predicts prognosis across gender and subtypes in papillary thyroid carcinoma. *Computational and Structural Biotechnology Journal*, 19, 1986-1997.

The valuable recommendations from the authors should be integrated/revised considering the following information from these recent publications regarding the thyroid-gut axis. Although no gut microbiota samples were collected and that relationship remains unexplored here, it's important to touch on these topics further:

- Nejman, D. et al. (2020). The human tumor microbiome is composed of tumor type-specific intracellular bacteria. *Science*, 368(6494), 973-980.

-Zhang, J., et al. (2019). Dysbiosis of the gut microbiome is associated with thyroid cancer and thyroid nodules and correlated with clinical index of thyroid function. *Endocrine*, 64(3), 564-574.

-Li, A. et al. (2021). Gut microbiome alterations in patients with thyroid nodules. *Frontiers in Cellular and Infection Microbiology*, 11, 643968-643968.

-Knezevic, J., et al. (2020). Thyroid-gut-axis: How does the microbiota influence thyroid function? *Nutrients*, 12(6), 1769.

-Ge, Y. et al. (2021). Gut microbiota influence tumor development and alter interactions with the human immune system. *Journal of Experimental & Clinical Cancer Research*, 40(1), 42-9.

Introduction: It's important to discuss that gut microbiota is comprised of not only bacteria but all organisms including fungi, protozoa, and viruses. Also, touching on the metabolites (e.g., short chain fatty acids) can be discussed, with relation to the immune system and how that may affect intra-tumor microbiota and thyroid cancer invasion. In line 26, it was briefly mentioned that germ-free rats have smaller thyroid glands. Some more information should be provided here to discuss why microbes are important for thyroid health as suggested. T4 and T3 are brought up with no introduction in line 31, so it would be nice to introduce them earlier in more detail. The same suggestion goes for Th1 and Th2 immune responses in that section. The characterization of the tumor microbiome has remained challenging because of its low biomass, which can be discussed further.

Methods: Please state your inclusion and exclusion criteria under the methodology section. The overall utilization of the ROC analysis can be optimized. Instead of using the three genera with higher abundance in T1 and T2, which may be causing over-fitting of the model, you can use random forest combined with a dimensionality reduction method (e.g., PCA), use cross-validation techniques, and then you will be able to list your top 10 genera of bacteria that are predictive of T1_T2 vs. T3_T4. That's a more interesting way of shaping the story. Also, please provide the p-value for your ROC curve. Please state your post hoc method in the methods section, as well as the bootstrapping which is mentioned under the figures.

OUT and OTU are used interchangeably throughout the paper.

Results: For Fig 4 please include what is G1 and G2. In line 153 of the results section, hormones are listed but haven't been described elsewhere. Please introduce these in the introduction section and share their un-abbreviated version.

Discussion: State the limitations of the study.

Reviewer #3 (Remarks to the Author):

The authors present the results of a study of the papillary thyroid cancer (PTC) intra-tumoral microbiota composition in 80 patients with localized PTC. They specifically describe the intra-tumoral microbiota composition using 16S rRNA sequencing of the V3-V4 region, observed across patients. They compare the microbiota composition and alpha-diversity based on PTC T-stage. They also describe the association between the intra-tumoral microbiota composition and thyroid hormone levels and markers of autoimmune thyroid disease. The authors find that intra-tumoral microbiota diversity (alpha-diversity) was significantly lower in patients with T1/T2 disease compared with T3/T4 disease using three distinct measures of alpha-diversity. Using assessment of beta-diversity, they show that tumor microbial communities in T1/T2 tumors have phylogenetic closeness to those in T3/T4 tumors. They describe the associations among intra-tumoral microbiota composition, alpha diversity, and several clinical parameters.

Overall Impression: Overall, the authors have a uniquely large cohort of patients with intra-tumoral microbiota sequencing. The intra-tumoral microbiota composition of PTC has not been well described previously, and the authors' finding of increased microbial diversity associating with increased tumor invasiveness (as measured by T stage) could carry potential future implications for screening/therapy. However, it is important to note that there are many flaws in this manuscript which would require revision prior to consideration of publication. Most notably, the authors over-state several conclusions as causative, when they are in fact only associations. There are numerous typos throughout the manuscript, which suggest that the authors have not proofread the work and many of the figures require revision for clarity and typos.

Specific Comments and Recommendations:

MAJOR:

1. Overstatement of conclusions: Authors state "Our finding suggested that the PTC tumor microbiome diversity and composition can influence the tumor invasion." – this is not correct, only

association was demonstrated. They similarly overstate causality later in the discussion, and all of these instances should be corrected.

2. Rationale for technique: The authors use a strategy of sequencing intra-tumoral bacteria. They should provide references to justify this technique, specifically that adequate bacterial sequences can be recovered using this strategy. In addition, would oral samples provide similar data? These would be easier to obtain and authors should consider whether future studies should include a comparison of intra-tumoral and oral microbiota.

3. Beta-diversity: This could be a valuable metric, not clearly explained in the paper. The authors state that the "overlap of OTUs from different groups was revealed, suggesting that the tumor microbial communities exhibited phylogenetic closeness." What they are really trying to explain is that there is phylogenetic/taxonomic overlap between T1-2 and T3-4 tumors.

4. Limitations: Authors need to expand upon the limitations of their approach.

5. Grammar/typos: Manuscript requires significant revision.

MINOR:

Title:

1. In the title, the last word ("invasion") should be capitalized if the remainder of the title is capitalized. In addition, the title should read either "Tumor Microbiome Diversity Influences Papillary Thyroid Cancer Invasion" or "Tumor Microbiome Diversity Influence on Papillary Thyroid Cancer Invasion"

Introduction:

2. The first paragraph of the introduction does not mention the microbiome at all – given that the manuscript is focused on the tumor microbiome/microenvironment, this should be mentioned in the first paragraph.

3. In referencing altered intra-tumoral microbiota observed across cancers, the authors fail to mention colon cancer, in which there have been many studies demonstrating the crucial role of the microbiome.

4. The authors interchange the use of the words "microbiota" and "microbiome" throughout the manuscript – microbiota refers to the actual organisms, while microbiome refers to the bacteria and their associated metabolome/microenvironment, so authors should be cautious in choosing the appropriate word.

5. Many typos:

- Line 4: should have a space between "the" and "endocrine"
- Line 5: missing the letter "d" on "solid tumor"
- Line 6: missing an "s" on "most patients"
- Line 23: left in the words "et al" after replacing reference with a number
- Line 33: "however characters of gut microorganisms in PTC remained explored" -> should possibly read "however, characteristics of gut microorganisms that play a role in PTC pathogenesis remain unexplored"?

Materials and Methods:

6. Lines 51-54: The authors should reference whether this technique has been used before to identify bacteria from tumor specimen. Do we have enough confidence from prior literature that this technique recovers bacteria from the tumor samples in a non-biased fashion?

7. The description of the sequencing methods is clear. The authors generated an OTU table, filtered out OTUs with low sequence frequency, and measured both alpha- and beta-diversity. The use of 16S rRNA sequencing with a focus on the V3-V4 region is appropriate.

8. Line 89: "a-diversity" should be "α-diversity"; should introduce the abbreviation "PCoA" when it is first mentioned

9. Line 94: "receiving operational curve" should be "receiver operating characteristic curve," or "ROC curve"

Results:

10. Patient characteristics: Table 1 formatting is poor. Authors should show the total number of patients included (n=80) at the top of the figure and show percentage of patients who were male (or female), percentage with each T stage. They should also provide rationale for showing T stage and nodal stage separately – is that common practice for PTC? If not, would be easier to visualize stage I, stage II, stage III etc. They do not need to show in the table that patients got chemoRT,

can simply state in the text that all patients received adjuvant chemoRT.

11. Tumor staging: as noted above, the authors focus on T stage, rather than overall stage – they should provide rationale for why they took this approach, not accounting for lymph node status in the analyses.

12. Alpha-diversity: The authors use both the Shannon and Simpson indices and sob as measures of alpha diversity. They should make note that the lower the value of the Simpson's index, the higher the diversity (opposite of Shannon index) – consider showing inverse Simpson's to make the message clearer. In addition, these concepts are shown in figure 1, but the figure is not referenced in the text. Authors should reference the figures.

a. If using all 3 methods of alpha diversity, authors should be able to comment on the different information provided by these 3 methods.

13. Beta-diversity: This could be a valuable metric, not clearly explained in the paper on lines 115-118. The authors state that the "overlap of OTUs from different groups was revealed, suggesting that the tumor microbial communities exhibited phylogenetic closeness." What they are really trying to explain is that there is phylogenetic/taxonomic overlap between T1-2 and T3-4 tumors. This is not necessarily unexpected given that these are all PTC samples. Perhaps of more interest would be the beta-diversity comparison of primary tumor and nodal metastasis. Otherwise, need to comment on a hypothesis – for example, maybe the authors hypothesized that they would see increased beta-diversity when comparing tumor samples of different T stages to explain different levels of invasiveness and that this was disproven/proven by the beta diversity analysis.

14. Authors should provide rationale for why they showed bacteria at the genus level in the paper rather than phylum level (shown in supplement).

15. Line 124: The authors state "At the genus level, Pseudomonas was considered as the clear advantage bacteria," – it is not clear what this means. Did authors mean the clearly "dominant" bacteria?

16. The information provided in figures 2D and 2E requires further explanation – it is not clear how this information is notably different from that in figures 2A-C.

17. Figure 2F is clear and provides interesting information about the different abundance of 3 key bacterial genus across T stage.

18. Figure 2G – it is very unclear what this is showing. Authors should provide further explanation about the components used to generate this ROC.

19. Thyroid hormones:

a. Line 151: Authors state "We investigated the effects of the thyroid hormones on intra-tumor microbiota in PTC" – not possible to show cause and effect in this type of analysis, should be investigated the correlation between thyroid hormones and microbiota

b. Seems like it would have been more useful to conduct this analysis on specific bacteria, rather than testing the association between thyroid markers and all of the genus shown – authors should comment specifically on whether they considered the effect of multiple hypothesis testing.

20. Clinical features

a. The authors test individual clinical features (age, gender) – should explain why they did not incorporate these into a multivariate analysis. We are less interested in microbiota association with each specific clinical feature and more interested in the microbiota and tumor invasiveness, controlling for effects of clinical variables.

Discussion:

21. In the discussion, should specify type of diversity that they are referring to since both alpha and beta diversity were discussed in the results.

22. Overstatement of conclusions: Authors state "Our finding suggested that the PTC tumor microbiome diversity and composition can influence the tumor invasion." – this is not correct, only association was demonstrated; also again misused "microbiome" vs. "microbiota"

Figures:

23. Figures 1 and 2: The text sizes are different across the figures on the Y axis which makes it harder to read

24. Figure 2: Would be helpful to show T stages in numerical order on the x-axis

25. Figure 2G: Typo in title – should read "ROC analysis on OTU level" (not "OUT")

26. Figure 4: Authors need to specify which gender is G1 and G2.

27. Figure captions: multiple typos with regard to grammar and OUT vs. OTU

Reviewers' comments:

Reviewer #1 (Remarks to the Author):

In the current manuscript, the authors present an analysis of the thyroid tumor microbiome and how it may affect carcinogenesis. The authors effectively present an interesting and concise conclusion of the relationship between the microbiome, TH, and antibodies and how they affect tumor invasion. However, there are some concerns with the justifications of certain methods and elaboration on their conclusions that must be addressed:

- The manuscript has formatting and grammatical errors. More thorough proofreading is necessary.

Response: The manuscript has been proofread thoroughly.

- It would be helpful to discuss examples of how microbes have shown to alter cancer progression or influence response to therapeutics in the Introduction. While the authors make it clear that the gut and intra-tumor microbiome are distinct in cancer samples, they should devote more time to explaining why and how microbes can be used as therapeutic agents.

It has been discussed in the “Introduction”

- There are different subtypes of PTC: classical, follicular variant, and tall cell. Each of these subtypes have shown to have different prognoses and response to treatment. Did the authors separate the tissue samples based on subtype? If not, why not?

It has been revised in Table 1, Figure S5-c, and “Results”, in line 170-173.

- Did the authors perform any procedures to correct for contamination? If not, the authors should discuss why not and how plausible/credible the results are without contamination correction.

It has been revised in the “Methods”; line 298-299. as follows:

“The negative control was subjected to PCR amplification to ensure that there was no contamination of the reagents.”

- Since the study primarily presents correlational data between microbe abundance and tumor invasion, thyroid hormones, and autoimmune antibodies, the authors should address how future research can confirm causality or the mechanism by which the microbes act (if such experiments are outside the scope of this study).

It has been discussed in the “Discussion,” line 243-245. as follows:

However, their roles in HT and GD remain to be explored. Furthermore, the mechanism by which the tumor microbiome modulates or resets the immune system may improve or impair the immune response against the tumor and affect PTC behavior and outcome.

- The author should not end the conclusion with an “In addition...” statement. The conclusion should do a better job in summarizing the study, synthesizing the results, and proposing next steps.

The “Discussion” has been revised thoroughly.

Reviewer #2 (Remarks to the Author):

In this study, the authors investigate how the tumor microbiome may determine tumor behaviour, affect patient outcome, and influence thyroid cancer. Interestingly, the intra-tumor dysbiosis, as well as its relationship with thyroid function, with regards to hormones and the immune system is also investigated. The topic is of interest to the readers of Communications Biology. Below are the reviewers’ comments that authors should address to improve the quality of the publication.

This is absolutely an important point worth studying, nonetheless, the following paper has been already published regarding intra-tumor microbiome and thyroid cancer. I think the authors should further stress the novelty of their approach against this paper:

- Gnanasekar, A. et al. (2021). The intratumor microbiome predicts prognosis across gender and subtypes in papillary thyroid carcinoma. Computational and Structural Biotechnology Journal, 19, 1986-1997.

It has been revised in the “Discussion” line 190-199, as follows:

A recent study demonstrated that the re-examination of whole-transcriptome RNA-sequencing (level 3 normalized mRNA expression read counts) and 450k DNA methylation sequencing data from the Genomic Data Commons legacy archive for microbial sequences enabled the identification of the association between different PTC subtypes and microbial abundance²⁸. Extending our analysis to the functional level demonstrated the correlation of certain immune or genetic traits and bacteria, despite a very large variation at the taxonomic level. In this study, using 16S sequencing methods, we validated the presence of bacteria in PTC and their function in tumor progression.

The valuable recommendations from the authors should be integrated/revised considering the following information from these recent publications regarding the thyroid-gut axis. Although no gut microbiome samples were collected and that relationship remains unexplored here, it’s important to touch on these topics further:

- Nejman, D. et al. (2020). The human tumor microbiome is composed of tumor type-specific intracellular bacteria. Science, 368(6494), 973-980.

-Zhang, J., et al. (2019). Dysbiosis of the gut microbiome is associated with thyroid cancer and thyroid nodules and correlated with clinical index of thyroid function. Endocrine, 64(3), 564-574.

-Li, A. et al. (2021). Gut microbiome alterations in patients with thyroid nodules. Frontiers in Cellular and Infection Microbiology, 11, 643968-643968.

-Knezevic, J., et al. (2020). Thyroid-gut-axis: How does the microbiome influence thyroid function? Nutrients, 12(6), 1769.

-Ge, Y. et al. (2021). Gut microbiome influence tumor development and alter interactions with the human immune system. Journal of Experimental & Clinical Cancer Research, 40(1), 42-9.

It has been revised in the manuscript, and all the publications have been cited in the manuscript.

Introduction: It’s important to discuss that gut microbiome is comprised of not only bacteria but all organisms including fungi, protozoa, and viruses. Also, touching on the metabolites (e.g., short

chain fatty acids) can be discussed, with relation to the immune system and how that may affect intra-tumor microbiome and thyroid cancer invasion. In line 26, it was briefly mentioned that germ-free rats have smaller thyroid glands. Some more information should be provided here to discuss why microbes are important for thyroid health as suggested. T4 and T3 are brought up with no introduction in line 31, so it would be nice to introduce them earlier in more detail. The same suggestion goes for Th1 and Th2 immune responses in that section. The characterization of the tumor microbiome has remained challenging because of its low biomass, which can be discussed further.

It has been revised in the “Introduction”; lines 32-35, 42-55, 57-64.

Methods: Please state your inclusion and exclusion criteria under the methodology section. The overall utilization of the ROC analysis can be optimized. Instead of using the three genera with higher abundance in T1 and T2, which may be causing over-fitting of the model, you can use random forest combined with a dimensionality reduction method (e.g., PCA), use cross-validation techniques, and then you will be able to list your top 10 genera of bacteria that are predictive of T1_T2 vs. T3_T4. That’s a more interesting way of shaping the story. Also, please provide the p-value for your ROC curve. Please state your post hoc method in the methods section, as well as the bootstrapping which is mentioned under the figures.

It has been revised in the “Method” line 330-338, as follows:

The differences between groups was analyzed by post-hoc test using Welch’s uncorrected test and adjusted by false discovery rate. The differences between groups were analyzed by the Wilcoxon rank-sum test using the two-tailed test. The 95% confidence interval was calculated using the bootstrap method (0.95). Circos analysis was used to describe the association between samples and species; LEfSe was used for linear discriminant analysis. To evaluate the discriminatory ability of the significant genus, a ROC curve was constructed, from which the AUC was calculated. The random forest model using cross-validation was also used to filtrate the significate genus, but lower AUC of 0.72 were obtain (Figure S3). The clinical variables were analyzed by chi-square using SPSS 24.0 (IBM Corp., Armonk, NY, USA).

OUT and OTU are used interchangeably throughout the paper.

It has been revised throughout the paper.

Results: For Fig 4 please include what is G1 and G2. In line 153 of the results section, hormones are listed but haven’t been described elsewhere. Please introduce these in the introduction section and share their un-abbreviated version.

It has been revised in the “Figure legend” line 502, “Introduction” line 51-55, and “Results” line 146-151.

Discussion: State the limitations of the study.

The limitations have been discussed in the “Discussion”; line 249-257.

Reviewer #3 (Remarks to the Author):

The authors present the results of a study of the papillary thyroid cancer (PTC) intra-tumoral microbiome composition in 80 patients with localized PTC. They specifically describe the intra-tumoral microbiome composition using 16S rRNA sequencing of the V3-V4 region, observed across patients. They compare the microbiome composition and alpha-diversity based on PTC T-stage. They also describe the association between the intra-tumoral microbiome composition and thyroid hormone levels and markers of autoimmune thyroid disease. The authors find that intra-tumoral microbiome diversity (alpha-diversity) was significantly lower in patients with T1/T2 disease compared with T3/T4 disease using three distinct measures of alpha-diversity. Using assessment of beta-diversity, they show that tumor microbial communities in T1/T2 tumors have phylogenetic closeness to those in T3/T4 tumors. They describe the associations among intra-tumoral microbiome composition, alpha diversity, and several clinical parameters.

Overall Impression: Overall, the authors have a uniquely large cohort of patients with intra-tumoral microbiome sequencing. The intra-tumoral microbiome composition of PTC has not been well described previously, and the authors' finding of increased microbial diversity associating with increased tumor invasiveness (as measured by T stage) could carry potential future implications for screening/therapy. However, it is important to note that there are many flaws in this manuscript which would require revision prior to consideration of publication. Most notably, the authors over-state several conclusions as causative, when they are in fact only associations. There are numerous typos throughout the manuscript, which suggest that the authors have not proofread the work and many of the figures require revision for clarity and typos.

The manuscript has been revised accordingly.

Specific Comments and Recommendations:

MAJOR:

1. Overstatement of conclusions: Authors state "Our finding suggested that the PTC tumor microbiome diversity and composition can influence the tumor invasion." – this is not correct, only association was demonstrated. They similarly overstate causality later in the discussion, and all of these instances should be corrected.

All the instances have been corrected.

2. Rationale for technique: The authors use a strategy of sequencing intra-tumoral bacteria. They should provide references to justify this technique, specifically that adequate bacterial sequences can be recovered using this strategy.

It has been revised in line 291-294, as follows:

The hypervariable regions V3-V4 of the bacterial 16S rRNA gene^{6, 7} were amplified with primer pairs 338F (5'-ACTCTACGGGAGGCAGCAG-3') and 806R (5'-GGACTACHVGGGTWTCTAAT-3') using an ABI GeneAmp[®] 9700 polymerase chain reaction (PCR) thermocycler (Applied Biosystems, CA, USA).

In addition, would oral samples provide similar data? These would be easier to obtain and authors should consider whether future studies should include a comparison of intra-tumoral and oral microbiome.

It has been discussed in line 254-257; as follows:

Moreover, oral microbiome also showed would exhibit potential association with cancer, and they are easier to obtain. Future studies including a comparison of intratumoral and oral microbiome may facilitate the characterization of the relationship between the tumor microbiome and tumor status further.

3. Beta-diversity: This could be a valuable metric, not clearly explained in the paper. The authors state that the “overlap of OTUs from different groups was revealed, suggesting that the tumor microbial communities exhibited phylogenetic closeness.” What they are really trying to explain is that there is phylogenetic/taxonomic overlap between T1-2 and T3-4 tumors.

It has been revised in line 87-91; as follows:

To gain a better understanding of the role of microbiome diversity, beta-diversity (β -diversity) was used to carry out a principal coordinate analysis (PCoA) using Bray–Curtis metric distances (Figure 1c). Significant differences in β -diversity were observed, further suggesting that the tumor microbial communities varied during tumor progression.

4. Limitations: Authors need to expand upon the limitations of their approach.

The limitations have been discussed in the discussion; line 249-257.

5. Grammar/typos: Manuscript requires significant revision.

The manuscript has been revised.

MINOR:

Title:

1. In the title, the last word (“invasion”) should be capitalized if the remainder of the title is capitalized. In addition, the title should read either “Tumor Microbiome Diversity Influences Papillary Thyroid Cancer Invasion” or “Tumor Microbiome Diversity Influence on Papillary Thyroid Cancer Invasion”

The title has been revised. Tumor Microbiome Diversity Influences Papillary Thyroid Cancer Invasion

Introduction:

2. The first paragraph of the introduction does not mention the microbiome at all – given that the manuscript is focused on the tumor microbiome/microenvironment, this should be mentioned in the first paragraph.

It has been revised in line 27-30; as follows:

Recently, the altered composition of intratumor microbiome has been observed in different cancers, which has been indicated to determine tumor behaviors and patient outcomes independent of the tumor genomic composition.

3. In referencing altered intra-tumoral microbiome observed across cancers, the authors fail to mention colon cancer, in which there have been many studies demonstrating the crucial role of the microbiome. microbiome

It has been revised in line 59-60. as follows:

However, the role of a specific intratumor microbiome is unclear because of its low biomass.

Intratumor microbiome has been revealed in cancers, such as head and neck squamous cell carcinoma²⁰, cholangiocarcinoma²¹, as well as colorectal²², pancreatic⁷, lung²³, urothelial²⁴, cervical²⁵, and breast cancer²⁶

4. The authors interchange the use of the words “microbiota” and “microbiome” throughout the manuscript – microbiota refers to the actual organisms, while microbiome refers to the bacteria and their associated metabolome/microenvironment, so authors should be cautious in choosing the appropriate word.

It has been revised as microbiome.

5. Many typos:

- Line 4: should have a space between “the” and “endocrine”
- Line 5: missing the letter “d” on “solid tumor”
- Line 6: missing an “s” on “most patients”
- Line 23: left in the words “et al” after replacing reference with a number
- Line 33: “however characters of gut microorganisms in PTC remained explored” -> should possibly read “however, characteristics of gut microorganisms that play a role in PTC pathogenesis remain unexplored”?

All the typos have been revised.

Materials and Methods:

6. Lines 51-54: The authors should reference whether this technique has been used before to identify bacteria from tumor specimen. Do we have enough confidence from prior literature that this technique recovers bacteria from the tumor samples in a non-biased fashion?

It has been revised in line 293, as follows:

The hypervariable regions V3-V4 of the bacterial 16S rRNA gene^{6, 7} were amplified with primer pairs 338F (5'-ACTCCTACGGGAGGCAGCAG-3') and 806R (5'-GGACTACHVGGGTWTCTAAT-3') using an ABI GeneAmp[®] 9700 polymerase chain reaction (PCR) thermocycler (Applied Biosystems, CA, USA).

7. The description of the sequencing methods is clear. The authors generated an OTU table, filtered out OTUs with low sequence frequency, and measured both alpha- and beta-diversity. The use of 16S rRNA sequencing with a focus on the V3-V4 region is appropriate.

8. Line 89: “a-diversity” should be “ α -diversity”; should introduce the abbreviation “PCoA” when it is first mentioned.

It has been revised in line 82, as follows:

The alpha-diversity (α -diversity),
and line 87 -90, as follows:

To gain a better understanding of the role of microbiome diversity, beta-diversity (β -diversity) was used to carry out a principal coordinate analysis (PCoA) using Bray–Curtis metric distances

9. Line 94: “receiving operational curve” should be “receiver operating characteristic curve,” or “ROC curve”.

It has been revised in line 125-126, as follows:

We then used the eight genera for the area under the curve (AUC)-receiver operating characteristic (ROC) analysis.

Results:

10. Patient characteristics: Table 1 formatting is poor. Authors should show the total number of patients included (n=80) at the top of the figure and show percentage of patients who were male (or female), percentage with each T stage. They should also provide rationale for showing T stage and nodal stage separately – is that common practice for PTC? If not, would be easier to visualize stage I, stage II, stage III etc. They do not need to show in the table that patients got chemoRT, can simply state in the text that all patients received adjuvant chemoRT.

It has been revised in Table 1, and line 279-283.

11. Tumor staging: as noted above, the authors focus on T stage, rather than overall stage – they should provide rationale for why they took this approach, not accounting for lymph node status in the analyses.

It has been revised in Table 1, and line 279-283.

12. Alpha-diversity: The authors use both the Shannon and Simpson indices and sob as measures of alpha diversity. They should make note that the lower the value of the Simpson's index, the higher the diversity (opposite of Shannon index) – consider showing inverse Simpson's to make the message clearer. In addition, these concepts are shown in figure 1, but the figure is not referenced in the text. Authors should reference the figures. a. If using all 3 methods of alpha diversity, authors should be able to comment on the different information provided by these 3 methods.

It has been revised in “figure legends” line 452-455, as follows:

“Note: The Sobs values indicate the number of observed OTUs, Shannon diversity index, and Simpson diversity index for an OTU definition, respectively. The lower the value of the Simpson index, the higher the diversity (opposite of the Shannon index).” and “results” line 85-87, The alpha-diversity (α -diversity) of the tumor microbiome was significantly lower in patients with T1/T2 PTC than in those with T3/T4 PTC, as shown by the Shannon and Simpson indices ($p=0.0309$ and $p=0.0088$; Wilcoxon rank-sum test was also performed Figure 1a). It indicated lower microbiome diversity (Shannon and Simpson indices) in patients with T1/T2 PTC. Microbiome richness was measured based on the number of observed OTUs (Sobs index), and no significant differences were found among different clinical stages (Figure 1b).

13. Beta-diversity: This could be a valuable metric, not clearly explained in the paper on lines 115-118. The authors state that the “overlap of OTUs from different groups was revealed, suggesting that the tumor microbial communities exhibited phylogenetic closeness.” What they are really trying to explain is that there is phylogenetic/taxonomic overlap between T1-2 and T3-4 tumors. This is not necessarily unexpected given that these are all PTC samples. Perhaps of more interest would be the beta-diversity comparison of primary tumor and nodal metastasis. Otherwise, need to comment on a hypothesis – for example, maybe the authors hypothesized that they would see increased beta-diversity when comparing tumor samples of different T stages to explain different levels of invasiveness and that this was disproven/proven by the beta diversity analysis.

It has been revised in lines 88-89, line 174-177.

14. Authors should provide rationale for why they showed bacteria at the genus level in the paper rather than phylum level (shown in supplement).

It has been revised in line 96-97, Figure 1, and Figure S1.

15. Line 124: The authors state “At the genus level, *Pseudomonas* was considered as the clear advantage bacteria,” – it is not clear what this means. Did authors mean the clearly “dominant” bacteria?

It has been revised in line 97-99. as follows:

At the genus level, *Pseudomonas* was the dominant bacteria, followed by *Rhodococcus*, *Ralstonia*, *Acinetobacter*, and *Sphingomonas*.

16. The information provided in figures 2D and 2E requires further explanation – it is not clear how this information is notably different from that in figures 2A-C.

It has been revised in line 102-105, “figure legend” line 468-473.

17. Figure 2F is clear and provides interesting information about the different abundance of 3 key bacterial genus across T stage.

18. Figure 2G – it is very unclear what this is showing. Authors should provide further explanation about the components used to generate this ROC.

It has been revised in line 125-133 and “figure legend” line 484-487.

19. Thyroid hormones:

a. Line 151: Authors state “We investigated the effects of the thyroid hormones on intra-tumor microbiome in PTC” – not possible to show cause and effect in this type of analysis, should be investigated the correlation between thyroid hormones and microbiome

It has been revised in line 137 and 138, 147-149.

b. Seems like it would have been more useful to conduct this analysis on specific bacteria, rather than testing the association between thyroid markers and all of the genus shown – authors should comment specifically on whether they considered the effect of multiple hypothesis testing.

It has been revised in lines 137 and 138, 153-154, and figure 3 and figure 3b.

20. Clinical features

a. The authors test individual clinical features (age, gender) – should explain why they did not incorporate these into a multivariate analysis. We are less interested in microbiome association with each specific clinical feature and more interested in the microbiome and tumor invasiveness, controlling for effects of clinical variables.

It has been revised in Table S1 and Table 1, in in line 253-256.

Discussion:

21. In the discussion, should specify type of diversity that they are referring to since both alpha and beta diversity were discussed in the results.

It has been revised in line 183-185: as follows:

We found that patients with advanced lesions (T3 or T4) had significantly higher tumor bacterial diversity (α -diversity) than those with relatively mild lesions (T1 or T2).

22. Overstatement of conclusions: Authors state “Our finding suggested that the PTC tumor microbiome diversity and composition can influence the tumor invasion.” – this is not correct, only association was demonstrated; also again misused “microbiota” vs. “microbiome”

It has been revised in line 261-262. In addition, microbiota and microbiome use has been reviewed throughout the text.

Figures:

23. Figures 1 and 2: The text sizes are different across the figures on the Y axis which makes it harder to read

24. Figure 2: Would be helpful to show T stages in numerical order on the x-axis

25. Figure 2G: Typo in title – should read “ROC analysis on OTU level” (not “OUT”)

26. Figure 4: Authors need to specify which gender is G1 and G2.

27. Figure captions: multiple typos with regard to grammar and OUT vs. OTU

All the figures have been revised.

Reviewers' comments:

Reviewer #1 (Remarks to the Author):

The authors used receiver operating characteristic analysis to identify an eight-genera microbiome signature that could reliably distinguish between different T stage. It is necessary for the authors to validate this signature on an external dataset or with a separate set of samples to confirm that this signature is generalizable.

The authors claim that PTC microbial host factors may determine the tumor behavior and patient outcome, and they discuss how specific microbes correlate with thyroid hormones and autoimmune thyroid disease-related antibodies. The study would be improved if the authors discussed what metabolites are produced by the specific microbes that were correlated with FT4, T4, FT3, and T3 and what pathways/factors these metabolites may regulate to disrupt thyroid hormone levels or to alter the structure of common chemotherapeutic agents. In addition to in vitro analysis, there are several bioinformatics pipelines that can provide metabolite and pathway information. This information is critical to provide more context to the microbes identified and truly elucidate their biological/clinical significance.

The authors wrote that "the negative control was subjected to PCR amplification to ensure that there was no contamination of the reagents." What exactly was amplified to show that the microbes in the tumor samples were truly differentially abundant and not a result of contamination by sequencing plate, sequencing date...etc. In a pan-cancer microbiome study published in Nature, Poore et al. used very stringent decontamination analyses that discarded up to 92.3% of total sequence data. Rigorous contamination correction is crucial for microbiome studies, so this point needs to be elucidated more clearly.

While the authors acknowledge several limitations of their study, their findings are not novel without addressing limitations about the generalizability and validity. At the very least, they must validate differential abundance of microbes and their correlations to thyroid hormones, autoimmune thyroid disease-related antibodies, and clinical variables in an external dataset. The authors should also add how their findings can be utilized for future research given the various limitations.

Reviewer #2 (Remarks to the Author):

The previous comments were addressed.

Some minor revisions can improve the quality:

Results section:

1. Line 87: please state the p-value for the b-diversity test.
2. Fig 1C-Please round off the p-values
3. Fig 2f, ensure the significance stars aren't overlapping the values
4. Fig 2i-Can you please provide a p-value for your AUC curve.
5. Fig 4b-Please round off the p-value
6. Lines 163-174: Please discuss figure 4b and provide the p-values within the text.
7. Fig S3,b-The placement for the AUC value should be consistent with other figures. Also please provide the p-values.
8. Fig S5-rounding the p-value.

Reviewer #3 (Remarks to the Author):

The authors have now satisfactorily addresses my concerns in this revised manuscript.

Reviewers' comments:

Reviewer #1 (Remarks to the Author):

The authors used receiver operating characteristic analysis to identify an eight-genera microbiome signature that could reliably distinguish between different T stage. It is necessary for the authors to validate this signature on an external dataset or with a separate set of samples to confirm that this signature is generalizable.

Reply: We revised Table S2, Figure S5-c, and the “Results” in lines 132-135 according to the reviewer’s comment.

The authors claim that PTC microbial host factors may determine the tumor behavior and patient outcome, and they discuss how specific microbes correlate with thyroid hormones and autoimmune thyroid disease-related antibodies. The study would be improved if the authors discussed what metabolites are produced by the specific microbes that were correlated with FT4, T4, FT3, and T3 and what pathways/factors these metabolites may regulate to disrupt thyroid hormone levels or to alter the structure of common chemotherapeutic agents. In addition to in vitro analysis, there are several bioinformatics pipelines that can provide metabolite and pathway information. This information is critical to provide more context to the microbes identified and truly elucidate their biological/clinical significance.

Reply: We revised Figures S2-c, S4-c,d, and S5-c,d and the “Results” in lines 141-148, 158-160, and 179-188 according to the reviewer’s comment.

The authors wrote that “the negative control was subjected to PCR amplification to ensure that there was no contamination of the reagents.” What exactly was amplified to show that the microbes in the tumor samples were truly differentially abundant and not a result of contamination by sequencing plate, sequencing date...etc. In a pan-cancer microbiome study published in Nature, Poore et al. used very stringent decontamination analyses that discarded up to 92.3% of total sequence data. Rigorous contamination correction is crucial for microbiome studies, so this point needs to be elucidated more clearly.

Reply: We revised the “Methods” in lines 330-332. The method used in the Nature study is 16S sequencing.

While the authors acknowledge several limitations of their study, their findings are not novel without addressing limitations about the generalizability and validity. At the very least, they must validate differential abundance of microbes and their correlations to thyroid hormones, autoimmune thyroid disease-related antibodies, and clinical variables in an external dataset. The authors should also add how their findings can be utilized for future research given the various limitations.

Reply: The limitations have been discussed in the discussion section; lines 280-299.

Reviewer #2 (Remarks to the Author):

The previous comments were addressed.

Some minor revisions can improve the quality:

Results section:

1. Line 87: please state the p-value for the b-diversity test.
2. Fig 1C-Please round off the p-values
3. Fig 2f, ensure the significance stars aren't overlapping the values
4. Fig 2i-Can you please provide a p-value for your AUC curve.
5. Fig 4b-Please round off the p-value
6. Lines 163-174: Please discuss figure 4b and provide the p-values within the text.
7. Fig S3, b-The placement for the AUC value should be consistent with other figures. Also please provide the p-values.
8. Fig S5-rounding the p-value.

Reply: All minor revisions have been performed in accordance with the reviewer's comment.

Reviewer #3 (Remarks to the Author):

The authors have now satisfactorily addresses my concerns in this revised manuscript.

Reply: We thank the reviewer for approving our manuscript.

REVIEWERS' COMMENTS:

Reviewer #1 (Remarks to the Author):

In this study, the authors investigating the intratumoral microbial landscape of PTC. After sequencing and performing correlational studies, they found specific microbes or bacteria taxa to correlate with specific clinical variables, hormones, and thyroid disease-related antibodies.

The authors have satisfactorily addressed my concerns in the revised manuscript.